# Sequences in the cytoplasmic tail of SARS-CoV-2 Spike facilitate expression at the cell surface and syncytia formation

Jérôme Cattin-Ortolá [1,2], Lawrence G. Welch [1,2], Sarah L. Maslen [1], Guido Papa[1], Leo C. James [1] & Sean Munro [1✉]

The Spike (S) protein of SARS-CoV-2 binds ACE2 to direct fusion with host cells. S comprises a large external domain, a transmembrane domain, and a short cytoplasmic tail. Understanding the intracellular trafficking of S is relevant to SARS-CoV-2 infection, and to vaccines expressing full-length S from mRNA or adenovirus vectors. Here we report a proteomic screen for cellular factors that interact with the cytoplasmic tail of S. We confirm interactions with the COPI and COPII vesicle coats, ERM family actin regulators, and the WIPI3 autophagy component. The COPII binding site promotes exit from the endoplasmic reticulum, and although binding to COPI should retain S in the early Golgi where viral budding occurs, there is a suboptimal histidine residue in the recognition motif. As a result, S leaks to the surface where it accumulates and can direct the formation of multinucleate syncytia. Thus, the trafficking signals in the tail of S indicate that syncytia play a role in the SARS-CoV-2 lifecycle.

[1] MRC Laboratory of Molecular Biology, Francis Crick Avenue, Cambridge CB2 0QH, UK. [2] These authors contributed equally: Jérôme Cattin-Ortolá, Lawrence G. Welch. ✉email: sean@mrc-lmb.cam.ac.uk

Coronavirus virions are encapsulated by a lipid bilayer that contains a small set of membrane proteins including the S protein that binds to, and fuses with, target cells, and the membrane (M) protein that recruits the viral genome into the virion[1–4]. Virions of SARS-CoV-2 and other betacoronviridae also contain the minor structural envelope (E) protein[5,6]. In infected cells, all three proteins are initially inserted into the endoplasmic reticulum (ER) and are then trafficked to the ER-Golgi intermediate compartment (ERGIC) and the Golgi where they are glycosylated, and in the case of S also cleaved, by Golgi enzymes[7,8]. Virions form by budding into the membranes of the ERGIC and Golgi, and membrane-bound carriers then transfer the newly-formed virions to the surface for release from the cell[9,10]. Once released, the virions thus display S on their surface where it can bind the host plasma membrane protein ACE2 to mediate fusion and thus infection of further cells[11–14]. Although SARS-CoV-2 and most other coronaviruses do not bud from the cell surface, some S protein is found on the surface of infected cells. Consistent with this, infected cells have been observed to fuse with neighbouring cells to form large multinucleate cells or syncytia[15–19].

Thus, S must exit the ER to reach the ERGIC and later Golgi compartments, with some travelling all the way to the cell surface but the majority being retained at the site of viral budding. In many coronaviruses, the latter retention in the Golgi is, in part, mediated by an interaction with M, which contains a Golgi localisation signal[2,20,21]. In addition, S proteins typically have binding sites for the COPI vesicle coat in their cytoplasmic tail, and it has been proposed that COPI-mediated recycling within the Golgi allows S to come into contact with M which then retains it there[20]. Insight into the intracellular traffic of S is thus relevant to understanding the replication of SARS-CoV-2 during infection. In addition, the majority of current SARS-CoV-2 vaccines are based on using either mRNA or adenovirus vectors to express the full-length S protein in cells of the vaccine recipient[22–25]. The location of S in the cell when expressed by the vaccine could affect its initial presentation to the immune system, and also the effect of the induced immunity on the S expressed by a subsequent vaccine boost.

S comprises a large external domain, a transmembrane domain (TMD) and a short cytoplasmic tail[26,27]. The traffic of membrane proteins through the secretory pathway is typically mediated by interactions between their cytoplasmic tails and the coat proteins that form the carriers that move proteins and lipids between compartments. In this work, we apply affinity chromatography to identify the cellular factors that recognise the cytoplasmic tail of S protein so as to investigate the mechanisms by which it is distributed between the membranes of host cells. By mapping the residues in the tail required for these interactions we then test their role in vivo, and so provide evidence that several features of the tail of SARS-CoV-2 enable its accumulation at the cell surface. These findings suggest that syncytia have a role in viral propagation, and also have implications for vaccine design.

## Results

**Affinity chromatography of cytosol with the cytoplasmic tail of S.** The 37 residue cytoplasmic tail of S was expressed in bacteria as a fusion to GST, and used for affinity chromatography of cytosol from human 293T cells. Proteins that bound preferentially to the tail fusion rather than GST alone were identified by mass spectrometry (Fig. 1a, b and Supplementary Data 1). The highest hits included several vesicle coat proteins and three related proteins, ezrin, moesin and radixin, that link membrane proteins to actin. Ezrin was previously reported as a hit in a yeast two-hybrid screen with the cytoplasmic tail of S from SARS, and suggested to

have a role in restraining viral entry[28]. The vesicle coat proteins include subunits of the coatomer complex that forms COPI-coated vesicles, consistent with a previous report that coatomer binds to the closely related cytoplasmic tail of the S protein of SARS[20]. In addition to these known interactors, there was a strong enrichment of subunits of the COPII coat that forms ER-to-Golgi vesicles, and of the sorting nexin SNX27 and its associated retromer complex that together recycle proteins from endosomes to the cell surface[29]. The few other strong hits are not linked to membrane traffic and were not investigated further, with the exception of WIPI3/WDR45B which interacts with membranes to regulate autophagy[30]. The significance of these other hits will thus not be clear until validated by in vitro and in vivo studies.

**Mapping of binding sites in the cytoplasmic tail of S.** To dissect the roles of the different coat proteins we mapped the regions that they bind on the 37 residue S protein tail (Fig. 1c, d). The tail comprises two distinct sections (Fig. 1e). The membrane-proximal half (1237–1254) contains eight cysteines which are known to be palmitoylated in the equivalent region in SARS and other coronaviruses and, once modified, are likely to be embedded in the surface of the bilayer[31,32]. The distal half of the tail (1255–1273) lacks cysteines and so will project into the cytoplasm. Testing GST fusions to these two halves showed that all the interactors bound to the distal region with the exception of SNX27 which exclusively bound to the cysteine-rich region (Fig. 1d, Supplementary Fig. 1a, b). To map binding at higher resolution, we tested tails with adjacent pairs of residues mutated to alanine (Fig. 1c). COPII binding was reduced by mutations in the acidic stretch DEDDSE that contains three copies of the di-acidic ER exit motif that is recognised by the Sec24 subunit of the COPII coat[33,34]. In contrast, COPI binding required the residues in the C-terminal KXHXX motif that was also found to be required for this interaction in SARS-CoV S protein[20]. Finally, binding of the FERM domain proteins required residues between the COPI and COPII binding sites, and SNX27 binding required residues in the N-terminal half of the tail nearest the TMD.

**Direct binding of trafficking proteins to the tail of S.** To further validate these interactions, we used recombinant proteins to test direct binding. The binding of COPI to KXKXX-type motifs is known to be primarily via the β-propellor domain of the β'-COP subunit[35,36], and we could confirm that this part of the human protein binds directly to the tail of S from SARS-CoV-2 (Fig. 2a). Recombinant SNX27 also bound directly to the tail, with residues 1238TheSerCys1240 next to the TMD being important, with the same residues found to be required for recruitment of SNX27 from cell lysate (Fig. 2b, Supplementary Fig. 1c). SNX27 associates with retromer via the latter's VPS26 subunit, and we found that VPS26 is recruited to the tail of S protein by the addition of SNX27 indicating that the tail can bind SNX27 whilst it is in a complex with retromer (Fig. 2c). Interestingly, some tail mutants that lost SNX27 binding retained binding to retromer indicating that this complex can also bind elsewhere in the tail. Mutation of residues required for SNX27 or moesin binding in full-length S protein did not detectably alter its intracellular distribution or accumulation on the plasma membrane (Supplementary Fig. 2a, b), indicating that these interactions do not have a role in cell surface delivery. Recently, two high-throughput screens for cellular factors required for SARS-CoV-2 infection of cultured cells identified retromer and SNX27 as hits, although this may be due to a role in the trafficking of ACE2[37,38]. It should also be noted that the binding of SNX27, although very efficient, is to the region

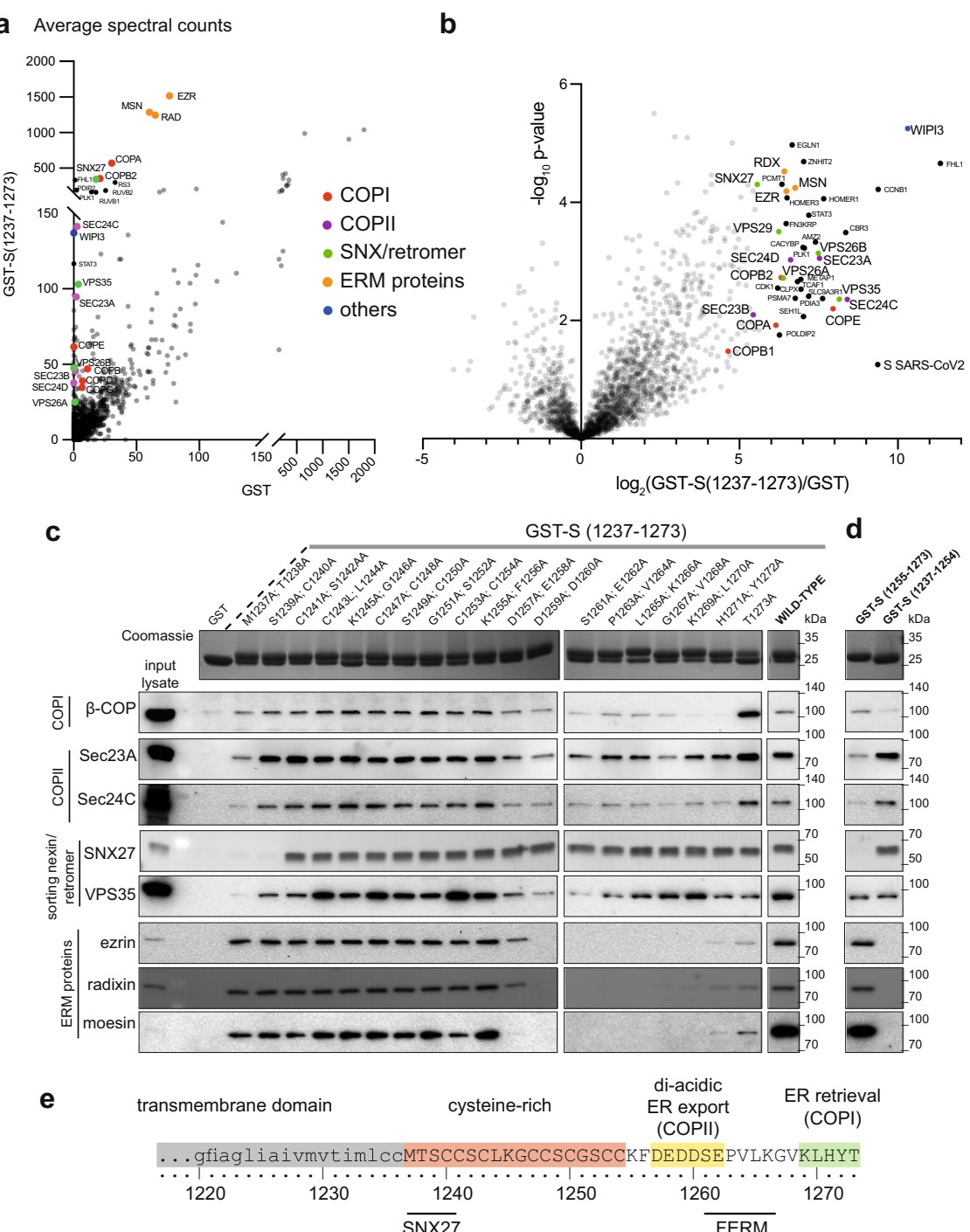

**Fig. 1 Proteomic analysis of the binding partners of the cytoplasmic tail of SARS-CoV-2 S protein. a** Mass spectrometry analysis from affinity chromatography of 293T cell lysates using GST-S tail (1237–1273). The plot compares the average spectral counts from three independent replicates of GST-S(1237–1273) versus the negative control (GST). Values are in Supplementary Data 1. **b** Volcano plot comparing the spectral intensities from proteins bound to GST-S(1237–1273) or GST alone, using data from three independent experiments. **c** Immunoblots of eluates from the indicated GST-tagged S tails prepared as in (**a**). Coomassie blue stained gels show the GST-tail fusions with adjacent residues of the tail of S (residues 1237–1273) mutated to alanine. The blots shown are representative from two independent experiments, and the input lysate represents 1/100 of the material applied to the GST fusions. **d** Validation of the interactions using two distinct halves of the tail of S: the membrane-proximal half (residues 1237–1254) and the distal half (residues 1255–1273). The blots shown are representative from three independent experiments. Binding of COPII is seen to residues 1237–1254 but this could be spurious—this region contains eight cysteines, and only one charged residue and so could be sticky in the absence of the rest of the tail and thus binds COPII non-specifically. In the context of the entire tail, none of the double alanine mutations in this region reduces substantially COPII binding, in contrast to what is seen with SNX27. **e** Schematic of the transmembrane and cytoplasmic domain of the SARS-CoV-2 S protein. The residues that are critical for binding to the different cytoplasmic factors are indicated.

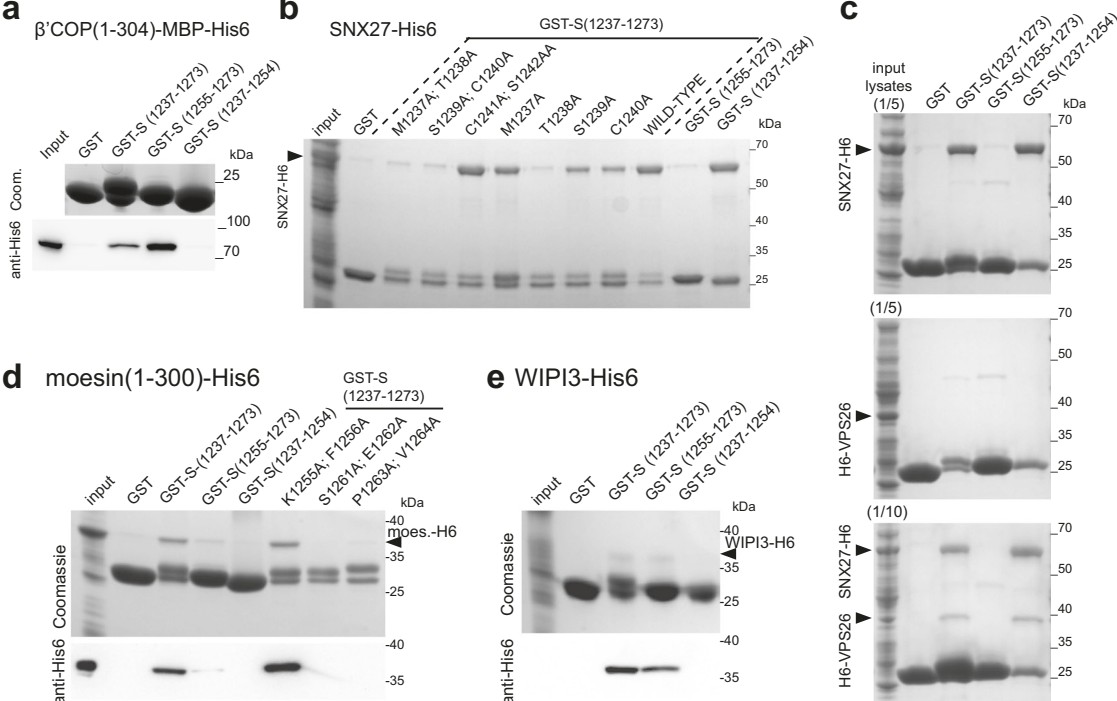

**Fig. 2 The tail of SARS-CoV-2 S protein binds directly to moesin, WIPI3, SNX27, and to VPS26 via SNX27. a** Coomassie-stained gel and immunoblot against His6 to test the binding from bacterial cell lysates of the β-propellor of the β'-COP subunit of COPI (residues 1–304 with an MBP-His6 tag) to beads coated with GST fusions to the tail of S. Experiment representative of three repeats with either His6-MBP-tag (as here) or with His6-tag. Input is 1/50 of that applied to beads. **b** Coomassie gel showing that the residues 1238ThrSerCys1240 of the tail of S are required for binding to SNX27. As in (**a**) except that the lysate was from cells overexpressing SNX27-His6. The experiment repeated three times, input 1/40 that applied to beads. **c** Coomassie gel showing that VPS26 requires SNX27 to bind to the tail of S protein. Same experiment as in (**a**) except that beads coated with the GST-fusions were incubated with bacterial lysate from cells overexpressing either SNX27-His6, His6-VPS26 or both. The experiment repeated twice. **d** Coomassie-stained gel and immunoblot against His6 to test the binding of the FERM domain of moesin (residues 1–300, expressed in bacterial lysate) to GST fusions to the tail of S. Experiment repeated twice, input is 1/40 (Coomassie) or 1/160 (immunoblot) of that applied to beads. **e** As in (**d**), except beads coated with the indicated GST-fusions were incubated with lysates from bacteria overexpressing WIPI3-His6. The experiment repeated twice, input is 1/14 (Coomassie) or 1/160 (immunoblot) of that applied to beads.

of the tail that will be palmitoylated in host cells and so the in vivo significance of this interaction remains uncertain.

Moesin is known to interact with plasma membrane proteins via its N-terminal FERM domain, and this part of the protein-bound directly to the S protein tail, with residues 1261SEPV being essential (Fig. 2d). The autophagy regulator WIPI3, when expressed in *E. coli*, also bound directly to the membrane distal half of the tail (Fig. 2e).

**Role of the binding sites for COPII in traffic of S**. We next examined the contribution of the COPI and COPII binding sites to the subcellular distribution of S protein. Mutation of the acidic residues in the COPII binding region greatly reduced cell surface expression, with S protein accumulating in the ER, indicating that these residues direct efficient egress of the newly made S protein out of the ER and into the secretory pathway (Fig. 3a–c).

**The binding site in S for COPI is sub-optimal**. The COPI binding region comprises KLHYT which differs somewhat from the canonical KXKXX or KKXX C-terminal COPI binding motif[35,39], and it has been speculated for SARS-CoV that this non-canonical motif might allow more S to reach the surface, although this was not tested experimentally[20]. We thus initially determined whether COPI binding to the KLHYT region was actually suboptimal by examining the effect on the binding of changing His1271 to a canonical lysine. For a H1271K variant of the tail, binding of COPI from cytosol was substantially increased,

whereas K1269A and H1271A variants showed reduced binding (Fig. 4a–c). Thus, the presence of His rather than Lys at residue 1271 results in the COPI binding site being suboptimal. In addition, in the above alanine scanning of the tail, mutation of the terminal residue Thr1273 to alanine was found to increase COPI binding (Fig. 1c), and this effect was recapitulated, suggesting the C-terminal threonine is a further feature of the tail that reduces its affinity for COPI.

Incorporation of the K1269A COPI binding site mutation into full-length S protein caused, at most, only a small increase in the cell surface expression (Fig. 4d). In contrast, the H1271K and T1273A mutations that increase COPI binding both caused S protein to instead accumulate intracellularly with substantial co-localisation with the ER (Fig. 4d, e). The proximity of the COPII and COPI binding sites in the very short tail of S means that mutations that affect the binding of one coat might also affect the other, especially as structures of COPI and COPII bound to peptides show that residues around the conserved motif can also make contact with the coat protein[35,36,40,41]. We thus also checked the binding of COPII to these variants, and found that indeed the H1271K mutation did reduce COPII binding slightly which may slow ER exit somewhat (Fig. 4b). However, the T1273A mutation did not bind less well to COPII and so the reduced cell surface expression is likely to reflect increased COPI binding rather than a reduction of interaction with COPII. Compared to H1271K, the T1273A mutant showed more post-ER staining in the perinuclear region, and this colocalised with the

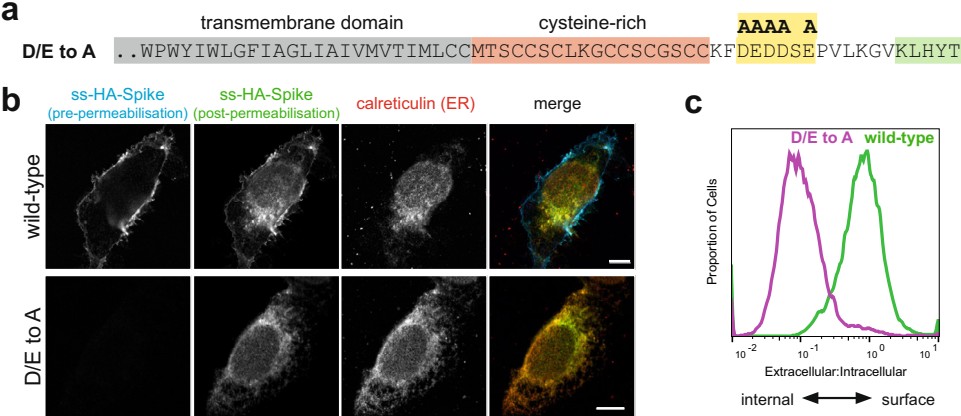

**Fig. 3 The cytoplasmic tail of the SARS-CoV-2 S protein harbours di-acidic ER export motifs. a** C-terminal sequence of the D/E to A mutant of S indicating the putative di-acidic ER export motifs mutated to alanine. **b** Micrographs of U2OS cells transiently expressing N-terminally HA-tagged wild-type S and the D/E to A mutant. Cell surface S was initially stained under non-permeabilising conditions with an anti-HA antibody labelled with Alexa Fluor (AF) 647. Cells were then permeabilised and stained with anti-HA labelled with AF488 to detect the internal S. Scale bars 10 μm, with the images for the individual channels taken at the same magnification as the merged image that has the scale bar. The experiment was repeated twice. **c** Quantification of (**b**) by flow cytometry. Displayed are overlaid histograms representing the ratio of extracellular S (AF488 signal) to that of intracellular S (AF647 signal) for wild-type S and the D/E to A mutant. Histograms normalised to the mode value, and represent >10,000 events. Chi-squared test shows the difference in the median ratios (wild-type, 0.78; D/E > A, 0,096) to be statistically significant ($P = 0.01$, 99% confidence). Representative of six independent experiments.

cis-Golgi marker GM130, consistent with a protein that is leaving the ER but then being recycled by binding COPI (Supplementary Fig. 2c).

These experiments were performed with S alone, but SARS-CoV-2 also expresses membrane (M) protein which in many coronaviruses, including SARS-CoV, is known to interact with S and cause it to accumulate in the Golgi where virion assembly occurs[2,20,21]. When the location of S was examined in cells co-expressing M, we found that, as expected, it caused S to accumulate with M in the Golgi, but it only reduced rather than eliminated the appearance of S on the surface (Supplementary Fig. 3a). This surface expression is further reduced by mutation of the COPII-binding site, and also by the H1271K and the T1273A mutations that optimise the COPI binding site (Supplementary Fig. 3b). Immunofluorescence shows that these variants still have the substantial ER staining seen in the absence of M (Supplementary Fig. 3c). In these cases, M does not accumulate in the ER as well, but rather is still localised to the Golgi. This is consistent with previous studies with other coronaviruses that found that S and M do not associate in the ER where they are both synthesised, but rather they only assemble after they have accumulated at the site of virion budding in the early Golgi[20,42,43]. Thus, even in the presence of M, the COPII binding site in S is required for exit from the ER, and an optimised COPI binding site in S can still be recognised so as to reduce transport to the surface.

**S is not efficiently endocytosed from the cell surface.** Thus, the COPI binding site in S has conserved features that reduce its ability to act as a retention signal in the Golgi and so allow at least some S to reach the cell surface. Some other coronaviruses, including those that have a histidine residue at the -3 position in the tail of S, have been found to be efficiently endocytosed if they reach the surface[44,45]. In these cases, endocytosis requires a tyrosine-containing motif that resembles the classic Yxxφ signal (Fig. 5a), but the S protein of SARS-CoV-2 lacks such a motif and consistent with this, we found that when it accumulated on the surface it was not efficiently endocytosed (Supplementary Fig. 2a, b).

**Syncytia formation is increased by the suboptimal COPI binding site.** What might be the reason for S protein to accumulate at the cell surface? SARS-CoV-2, like other coronaviruses, buds into intracellular membranes and so S protein that has reached the plasma membrane will not contribute to virion formation but it is in a position to cause infected cells to fuse to adjacent cells and so facilitate spread without virion release. We thus tested the effect of the mutations in the COPI binding site on the degree of cell fusion induced by S protein. 293T cells were transfected with a plasmid expressing S protein and mixed with Vero cells expressing human ACE2. In this assay system S is expressed in cells that are human but do not express ACE2, and the fusion target expressed human ACE2, and is a cell line widely used for fusion with S or intact virus, with fusion being followed in this case by mixing of different fluorescent markers present in the two cell types[46]. Using this assay we found that the K1269A mutant that prevents COPI binding caused a small but reproducible increase in cell fusion (Fig. 5b, c). In contrast, the H1271K mutation that binds better to COPI resulted in greatly reduced cell fusion, with the mutant S also showing reduced levels of S1/S2 cleavage, consistent with it not moving beyond the early Golgi (Fig. 4d). It should be noted that this assay was performed using, as the fusion target, monkey (Vero) cells overexpressing human ACE2. This will increase the susceptibility of the cells to S-mediated fusion, and hence it is striking that the H1271K mutant still reduces fusion, but it also means that caution is needed in extrapolating these findings to other cell types. Nonetheless, the results clearly indicate that, at least in this context, the sub-optimal COPI binding site enhances the ability of S to form syncytia.

**Discussion**

The data presented here indicate that the S protein of SARS-CoV-2 has three features that facilitate its accumulation on the plasma membrane. Firstly, a region containing di-acidic COPII binding motifs directs efficient exit from the ER. Secondly, the COPI-binding site is suboptimal which allows some S protein to escape the Golgi apparatus. Consistent with this, the S protein of the coronavirus porcine epidemic diarrhea virus (PEDV) has a

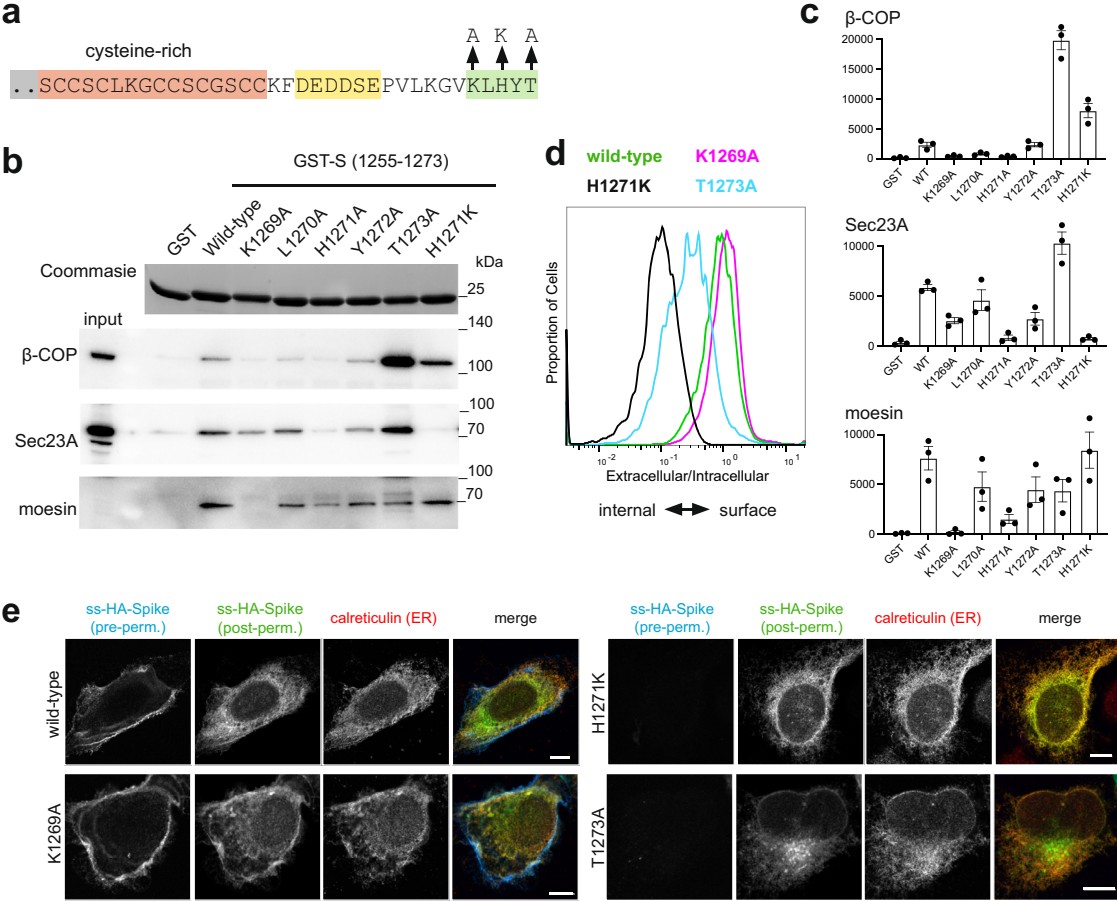

**Fig. 4 The cytoplasmic tail of the SARS-CoV-2 S protein has a suboptimal ER retrieval motif. a** Schematic of the cytoplasmic tail of S protein showing three different mutations made in the COPI-binding site. **b** Beads coated with the indicated recombinant S tail GST-fusions were incubated with clarified lysate from 293T cells and the eluates immunoblotted for β-COP (COPI), Sec23A (COPII) and moesin. Representative blots from three independent experiments, input 1/50 of that applied to beads. **c** Quantification of the blots in (b), data from the three independent experiments (arbitrary units), showing mean values and SEMs. **d** Flow cytometry analysis of U2OS cells expressing the indicated forms of S. The histograms represent the ratio of extracellular S (AF488 signal) to that of intracellular S (AF647 signal). Histograms normalised to the mode value and represent ~10,000 events. Chi-squared tests show the differences between the median ratios of the wild-type (0.87) and those of K1269A (1.1), H1271K (0.11), and T1273A (0.12) to all be statistically significant (*P* = 0.01, 99% confidence). Representative of four independent experiments. **e** Micrographs of U2OS cells transiently expressing N-terminally HA-tagged wild-type S or the indicated mutants. Cell surface S was initially stained using AF647-labelled anti-HA under non-permeabilising conditions. Cells were then permeabilised and stained with AF488-labelled anti-HA to detect the internal S, and for the ER marker calreticulin. Both H1271K and T1273A accumulate in the ER, with the latter's additional perinuclear location presumably reflecting subtle differences in relative ER-to-Golgi, and Golgi-to-ER transport kinetics due to differential effects on COPI and COPII binding. Scale bars 10 μm, with the images for the individual channels taken at the same magnification as the merged image that has the scale bar. The experiment was repeated twice.

related C-terminal sequence (-KVHVQ) and binds COPI with a much lower affinity than canonical KXKXX motifs[36]. Finally, the S-protein of SARS-CoV-2 is not efficiently endocytosed, consistent with it lacking a tyrosine-containing motif of the sort that is found in many other coronaviruses, including PEDV (Fig. 5a). For several of these other coronaviruses the tyrosine-containing motifs have been shown to either induce endocytosis or prevent movement beyond the Golgi[44,45,47,48]. Our data do not provide insight into possible roles for the other interactions that we observed, and progress may require testing in different cells or in conditions of virus replication. Nonetheless, it is worth noting that the ERM proteins anchor plasma membrane proteins to the actin cytoskeleton and hence can direct their accumulation at specific regions of the cell surface, and also that cell fusion proteins expressed by reoviruses have been found to direct actin filament formation to drive cell fusion[49,50]. Two recent studies have reported interaction screens of all proteins expressed by SARS-CoV-2, looking for host proteins that co-precipitate with

each viral protein when expressed individually in cultured cells[51,52]. In both cases, the only interaction partners identified for S were the palmitoyltransferase subunits ZDHHC5 and GOLGA7. However, in both studies, S was expressed with an HA-tag at the C-terminus of the cytoplasmic tail. Not only does the KXKXX motif have to be at the C-terminus for recognition by COPI, but it also seems likely that the presence of a bound antibody adjacent to the small cytoplasmic tail would obstruct the binding of large coat protein complexes[39].

Overall, our results, along with previous studies on SARS-CoV, provide a clear explanation of why S accumulates on the cell surface when expressed alone. In SARS-CoV-2 infected cells the majority of S accumulates at the site of virion biogenesis in the early Golgi due to an interaction with M, as happens with other coronaviruses including SARS[2,20,53]. Nonetheless, even in SARS-CoV-2 infected cells, it is clear that at least some S reaches the surface, and syncytia formation has been observed in cultured airway epithelial cells infected with SARS-CoV-2 in vitro, in post

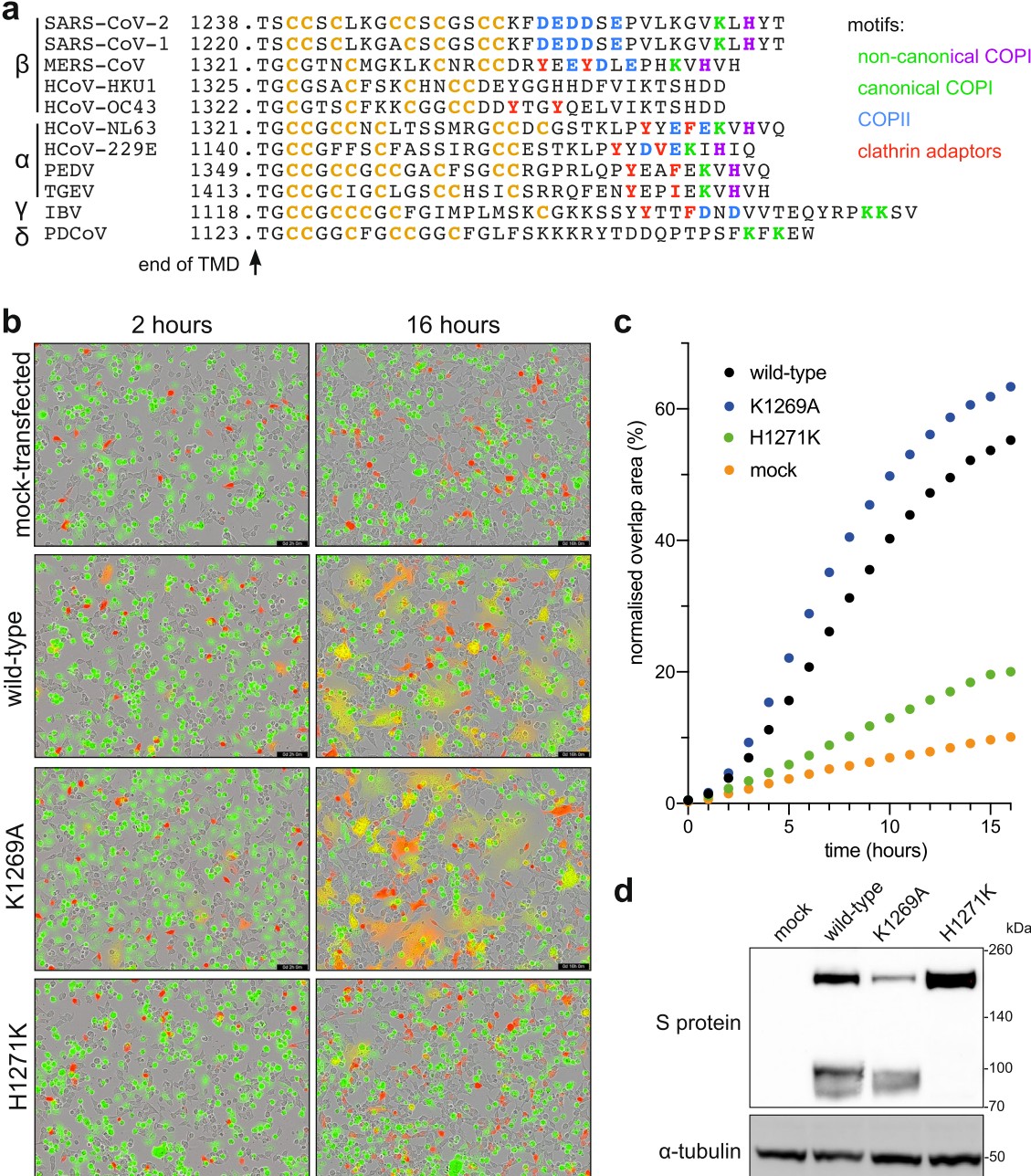

**Fig. 5 SARS-CoV-2 S leakage to the plasma membrane facilitates cell-cell fusion and the formation of syncytia. a** Comparison of the cytoplasmic tails of S proteins from the four genera of Coronaviridae, aligned by their TMDs. Binding motifs for trafficking proteins are indicated, along with cysteine residues (orange). **b** Representative micrographs showing 293T cells coexpressing mCherry and different full-length S variants fusing with acceptor Vero cells stained with CellTracker Green. Cell fusion is monitored by overlap between the green and red signal. Retention of S in the ER reduces its capacity to form syncytia while increased leakage to the plasma membrane increases it. Scale bars: 200 μm. **c** Quantification of cell-cell fusion as percentage of the field showing green and red overlap. Mean of two biological replicates (see Source Data file). **d** Immunoblots from 293T cells expressing the indicated full-length S variants, with the H1271K mutation preventing the protein from reaching the site of S1/S2 cleavage by furin in the late Golgi. Representative of two independent replicates.

mortem samples of patients who have died of COVID-19, and in nonhuman primate models[16,17,54,55]. Syncytia formation is a known feature of a diverse range of viruses, including other respiratory viruses such as measles and respiratory syncytial virus. It is proposed that it allows the rapid and efficient spread of viruses between cells in a manner that evades immune surveillance[56,57]. One striking example is the reoviruses which are not enveloped, but particular strains have been found to carry a gene for a fusion protein that directs cell-cell fusion, with this

fusion protein being required for pathogenicity but not replication[58,59]. Indeed, it seems conceivable that the formation of large syncytia could increase viral pathogenicity by destablising airway epithelia or creating holes that are harder to repair than the loss of a single infected cell. It has recently been reported that syncytia formation by SARS-CoV-2 S can induce pyroptosis, and it has been proposed for other viral-mediated fusion events that cell death following syncytia formation can affect immune responses[60,61]. All this has led to an interest in the possibility that

syncytia formation by SARS-CoV-2 is a potential target for therapeutic strategies[18,62]. Indeed, the process appears entirely dependent on the cell surface protease TMPRSS2 to make the activating S2' cleavage, whereas viral entry can be facilitated by either TMPRSS2 or lysosomal cathepsins[11,17]. Nonetheless, it should be stressed that for SARS-CoV-2 the importance of syncytia for either infection or pathogenicity remains to be proven, although our findings at least suggest a means by which this could be tested by examining the effect of making changes in the cytoplasmic tail that reduce the cell surface expression of S, and hence the rate of syncytia formation, without otherwise affecting viral replication.

The trafficking of S protein to the cell surface has implications beyond understanding SARS-CoV-2 infection, as intact S is also expressed by the current vaccines based on mRNA or adenovirus vectors[22–25]. In this case, S is expressed without other viral proteins and so it will almost certainly to be efficiently trafficked to the cell surface. Syncytia formation is unlikely to be an issue as most of the vaccines express a version of S that is stabilised in the pre-fusion state by changing two residues to proline, and this change greatly reduces syncytia formation[22,63]. However, the location of S within the cell could affect the efficiency with which it is recognised as a non-host protein and then processed by the immune system to generate an immune response. Once an immune response has been evoked, a second vaccination will expose more S on the surface of cells, and again, altering the location of S could conceivably reduce adverse reactions from immune attack on host cells, especially if multiple rounds of vaccination prove to be required to contain future variants of SARS-CoV-2.

Clearly, further studies will be required for investigating the importance of the intracellular location of S for both SARS-CoV-2 infection and vaccine design. However, our studies should provide a framework for testing the effect on both processes of altering the location of S by rational mutation of the trafficking signals in its cytoplasmic tail.

## Methods

**Plasmids.** Details of the plasmids and relevant primers used in this report can be accessed from Supplementary Data 2. Constructs used for bacterial expression of the cytoplasmic domains of the Spike protein (S) of SARS-CoV-2: the sequence encoding the N-terminally GST-tagged cytoplasmic domain of S were cloned into the vector pGEX6p2 (GE Healthcare Life Sciences). For GST-S (1255-1273, pJC149) and GST-S (1237-1273, pJC150), the inserts and vector were amplified by PCR and assembled using Gibson cloning with the linker SDLEVLFQGPLGSP-GIQ. For GST-S (1237–1254, pJC247) in pGEX6p2, sequence-specific primers were annealed and inserted at the EcoRI and BamHI sites with the linker SDLEVLFQGPLGSPGIQ. For the alanine mutants of GST-S (pJC173 to pJC190, and pJC254 to pJC257) two fragments of pJC150 were amplified by PCR using mutation-specific primers and assembled by Gibson cloning. gBlocks and PCR amplified vector were assembled by Gibson cloning. or the constructs used in Fig. 4b to express single alanine mutations pLGW829, and pJC206 to pJC210, mutation-specific oligos were annealed and ligated to pGEX6p2 digested with EcoRI and XhoI.

Constructs used for bacterial expression of the described S protein interactors: full-length human WIPI3/WDR45B (NCBI: NM_019613.4, UniProt Q5MNZ6) expressed in pOPT vector (from O. Perisic, MRC-LMB) with a C-terminal PGAGA linker and a His6 tag was generated through Gibson assembly of the PCR amplified insert (pJC248 was the template) and vector. Coding regions for moesin (1–300)-His6 (UniProt P26038), and human SNX27-His6 (528 amino acid version UniProt Q96L92-3) were cloned into pOPT vector with a C-terminal PGAGA linker and a His6 tag. Inserts were synthesised as gene fragments (IDT for moesin [1–300] and Genewiz for SNX27) with codons optimised for E. coli expression. gBlocks and PCR amplified vectors were assembled by Gibson cloning. Coding regions for human COPB2 (1–304) in pOPT vector with a C-terminal PGAGA linker and either a His6 or an MBP-His6 tag were generated through Gibson assembly of an insert synthesised gene fragment (codon optimised for E. coli expression) and the PCR amplified vector.

WIPI3-HA construct used for expression in mammalian cells (pJC248): a gBlock (IDT) encoding WIPI3-HA was assembled by Gibson cloning with PCR amplified pcDNA3.1+ (Clontech). ss-HA-CPD in pEGFP_N1 (pJC338): signal sequence (1–31) with the HA tag in the reverse primer, carboxypeptidase D (CPD)

(32–1380) and vector were PCR amplified and the three fragments were assembled by Gibson cloning. CPD cDNA was from GenScript clone ID OHu10876, accession number NM_001304.4.

Full length S protein constructs used for expression in mammalian cells: a sequence encoding SARS-CoV-2 S was codon optimised for expression in mammalian cells and cloned into pcDNA3.1+ (modified to be compatible with the PiggyBac transposase system) using the restriction sites NheI and NotI. Where indicated, an HA tag was inserted after the signal peptide by introduction into the forward primer, amplification by PCR and insertion into pcDNA3.1+. Key residues in the cytoplasmic tail of S were mutated as indicated by introducing mutations into primers, amplification of a small region at the 3' end of the gene and insertion using the restriction sites BstEII and NotI.

**Mammalian cell culture.** Human embryonic kidney 293T (ATCC, CRL-3216), U2OS (ATCC, HTB-96) and Vero (ATCC, CCL-81) cells were cultured in Dulbecco's modified Eagle's medium Glutamax (DMEM; Gibco) supplemented with 10% fetal calf serum (FCS) and penicillin/streptomycin at 37 °C and 5% $CO_2$. Unless indicated otherwise, cells were transfected using polyethylenimine (PEI; Polyscience, 24765) dissolved in PBS to 1 mg/ml. The ratio of PEI (μL) to DNA (μg) used was 3:1; PEI was dissolved in Opti-Mem, incubated at room temperature for 5 min, DNA was added and incubated for another 15–20 min at room temperature before dropwise addition onto cells which had been seeded the day before. Cells were not authenticated after being obtained from the ATCC, but were free of mycoplasma as determined by regular testing (MycoAlert, Lonza).

**Protein expression in bacteria.** GST (pGEX6p2 vector), the different versions of GST-S constructs and the His6-tagged interactors were expressed as follows: plasmids were transformed into E. coli BL21-CodonPlus (DE3)-RIL (Agilent, 230245). From an overnight starter culture, cells were grown in 2xTY medium containing 100 μg/mL ampicillin (or 50 μg/mL Kanamycin for His6-VPS26) and 34 μg/mL chloramphenicol at 37 °C in a shaking incubator. When the culture reached $OD_{600} = 0.6 - 0.8$, the temperature was lowered to 16 °C, protein expression was induced with 100 μM of Isopropyl β-D-1-thiogalactopyranoside (IPTG), and incubated overnight. Bacteria cells were harvested by centrifugation at $4000 \times g$ at 4 °C for 15 min and were mechanically resuspended on ice in lysis buffer containing 50 mM Tris, pH 7.4, 150 mM NaCl, 1 mM EDTA, 5 mM 2-mercaptoethanol, 1% Triton X-100, and supplemented with protease inhibitor cocktail (cOmplete, Roche). Cells were lysed by sonication and the lysates were clarified by centrifugation at $20,000 \times g$ at 4 °C for 15 min. Clarified lysates were flash frozen in liquid nitrogen and thawed as needed for the binding assays.

**GST-pulldowns using 293T cell lysates.** Pull downs for mass spectrometry: clarified lysates from 450 mL 2xTY cultures containing bacteria expressing recombinant GST, GST-S tails (product of pJC149, pJC150, pJC247) were thawed. 100 μL of glutathione Sepharose 4B bead slurry (GE17-0756-01) was washed twice with lysis buffer (50 mM Tris, pH 7.4, 150 mM NaCl, 1 mM EDTA, 5 mM 2-mercaptoethanol, 1% Triton X-100) by centrifugation at $100 \times g$ for 1 min at 4 °C and aspiration of the washing buffer. Clarified bacterial lysates were added to the glutathione Sepharose beads, and incubated at 4 °C for 1 h on a tube roller. 293 T cells (from four confluent T175 flasks per GST-tagged bait) were collected by scraping and lysed with lysis buffer supplemented with protease inhibitor cocktail (EDTA-free, cOmplete, Roche). The lysate was clarified by centrifugation for 5 min at $17,000 \times g$ and pre-cleared with 100 μL of Glutathione Sepharose bead slurry per bait. Beads loaded with recombinant GST-tagged baits were washed once with ice-cold lysis buffer, once with lysis buffer supplemented with 500 mM NaCl, and once again with lysis buffer. Around 5% of the beads were kept aside as an input control and the remaining beads were incubated with the pre-cleared 293T cell lysate for 2–4 h on a tube roller at 4 °C. Beads were washed twice with lysis buffer, transferred to 0.8 mL centrifuge columns (Pierce 89869B) and washed twice more. Columns were brought to room temperature and eluted 5 times with 100 μL of elution buffer (1.5 M NaCl in lysis buffer) by centrifugation at $100 \times g$ for 1 min; for the final elution the sample was centrifuged at $17,000 \times g$ for 1 min. Eluates were pooled together and concentrated down to around 75 μL using an Amicon Ultra 0.5 mL 3000 NMWL centrifugal filter (Millipore UFC500324), supplemented with 25 μL of NuPage 4x LDS sample buffer (Invitrogen, NP0007) containing 100 mM DTT. 40% of the eluate was loaded on SDS PAGE gels (Invitrogen, XP04202) and stained with InstantBlue Coomassie Protein Stain (Abcam, ab119211). Each lane was cut into 8 gel slices for mass spectrometry analysis.

Pulldowns for western blotting: pulldowns were conducted as described for mass spectrometry with the exception that half of the amounts of clarified bacterial lysate, Glutathione Sepharose beads, and clarified 293T cell lysate were used. The amounts of eluates indicated on the figures were loaded on to SDS PAGE gels (Invitrogen, XP04205). GST-pulldown with 293T cells expressing WIPI3-HA: cells were transiently transfected with WIPI3-HA (pJC248). Forty-eight hours later, cells were collected by scraping, lysed and used for the pulldown as described above with the exception that for each bait, cells from half of a T175 flask were used. For the GST-pulldowns in Fig. 4b, 50 mM HEPES pH 7.4 was used instead of 50 mM Tris pH 7.4.

**Mass spectrometry**. Polyacrylamide gel slices (1–2 mm) were placed in a well of a 96-well microtitre plate and destained with 50% v/v acetonitrile and 50 mM ammonium bicarbonate, reduced with 10 mM DTT, and alkylated with 55 mM iodoacetamide. After alkylation, proteins were digested with 6 ng/μL trypsin (Promega, UK) overnight at 37 °C. The resulting peptides were extracted in 2% v/v formic acid, 2% v/v acetonitrile. The digests were analysed by nano-scale capillary LC-MS/MS using an Ultimate U3000 HPLC (ThermoScientific Dionex, San Jose, USA) to deliver a flow of ~300 nL/min. A C18 Acclaim PepMap100 5 μm, 100 μm × 20 mm nanoViper (ThermoScientific Dionex, San Jose, USA), trapped the peptides prior to separation on a C18 BEH130 1.7 μm, 75 μm × 250 mm analytical UPLC column (Waters, UK). Peptides were eluted with a 60-minute gradient of acetonitrile (2–80%). The analytical column outlet was directly interfaced via a nano-flow electrospray ionisation source, with a quadrupole Orbitrap mass spectrometer (Q-Exactive HFX, ThermoScientific, USA). MS data were acquired in data-dependent mode using a top 10 method, where ions with a precursor charge state of 1+ were excluded. High-resolution full scans ($R = 60,000$, $m/z$ 300–1800) were recorded in the Orbitrap followed by higher energy collision dissociation (HCD) (26% Normalised Collision Energy) of the 10 most intense MS peaks. The fragment ion spectra were acquired at a resolution of 15,000 and dynamic exclusion window of 20 s was applied. Raw data files from LC-MS/MS data were processed using Proteome Discoverer v2.1 (Thermo Scientific), and then searched against a human protein database (UniProtKB, reviewed) using the Mascot search engine programme (Matrix Science, UK).

Database search parameters were set with a precursor tolerance of 10 ppm and a fragment ion mass tolerance of 0.2 Da. One missed enzyme cleavage was allowed and variable modifications for oxidised methionine, carbamidomethyl cysteine, pyroglutamic acid, phosphorylated serine, threonine and tyrosine were included. Total spectral counts were determined with the protein threshold set at 80%, the minimum number of peptides set as 2 and the peptide threshold set at 50%.

**Analysis of mass spectral intensities**. All raw files were processed with Max-Quant v1.5.5.1 using standard settings and searched against the UniProt Human Reviewed KB with the Andromeda search engine integrated into the MaxQuant software suite[65,66]. Enzyme search specificity was Trypsin/P for both endoproteinases. Up to two missed cleavages for each peptide were allowed. Carbamido-methylation of cysteines was set as fixed modification with oxidised methionine and protein N-acetylation considered as variable modifications. The search was performed with an initial mass tolerance of 6 ppm for the precursor ion and 0.5 Da for MS/MS spectra. The false discovery rate was fixed at 1% at the peptide and protein level. Statistical analysis was carried out using the Perseus module of MaxQuant[67]. Prior to statistical analysis, peptides mapped to known contaminants, reverse hits and protein groups only identified by site were removed. Only protein groups identified with at least two peptides, one of which was unique and two quantitation events were considered for data analysis. Each protein had to be detected in at least two out of the three replicates. Missing values were imputed by values simulating noise using the Perseus default settings. To calculate the p-values, two sample t-tests were performed.

**Immunoblotting**. Protein samples in 1x NuPage LDS sample buffer, 25 mM DTT (or 50 mM TCEP pH 7.0) were loaded on to SDS PAGE gels (Invitrogen, XP04205) and transferred to nitrocellulose membranes. Membranes were blocked in 5% (w/v) milk in PBS-T (PBS with 0.1% [v/v] Tween-20) for 1 h, incubated overnight at 4 °C with the primary antibody in the same blocking solution, washed three times with PBS-T for 5 min, incubated with HRP- or Alexa Fluor (AF)-conjugated secondary antibody (where indicated) in 0.1% (w/v) milk in PBS-T for 1 h and, washed three times with PBS-T for 5 min. Where indicated a primary anti-His6 or anti-SNX27 antibody crosslinked to HRP was used. Blots stained with HRP-conjugated secondary antibodies were imaged using BioRad ChemiDoc imagers with chemiluminescence substrates: either SuperSignal West Pico PLUS (Thermo Scientific, 34577) or SuperSignal West Femto (Thermo Scientific, 34095) depending on the strength of the signal. Blots stained with AF-conjugated secondary antibodies were imaged using a Typhoon Imager (GE Healthcare) or a BioRad ChemiDoc. Primary and secondary antibodies, their catalog numbers, and their dilutions for use are in Supplementary Data 2. The gels analysis tool in ImageJ was used for the quantification in Fig. 4c.

**In vitro binding assays using recombinant proteins**. Saturating amounts of clarified bacterial lysates containing GST and GST-S fusions were added to Glutathione Sepharose beads that were previously washed with lysis buffer (50 mM Tris, pH 7.4, 150 mM NaCl, 1 mM EDTA, 5 mM 2-mercaptoethanol, 1% Triton X-100) and incubated at 4 °C for 1 h on a tube roller. Beads were washed once with lysis buffer, once with lysis buffer supplemented with 500 mM NaCl, once again with lysis buffer, and incubated with clarified bacterial lysates containing recombinant WIPI3-His6, moesin(1–300)-His6, COPB2(1–304)-MBP-His6, SNX27-His6, or His6-VPS26 for 2 h at 4 °C on a rotator. Beads were washed three times with lysis buffer and eluted with 25 mM reduced glutathione made in lysis buffer, supplemented with a 4× solution of NuPage LDS sample buffer containing 100 mM DTT and loaded on SDS PAGE gels (Invitrogen, XP04205) and analysed by Coomassie protein stain (Abcam, ab119211) or by immunoblot using an anti-His6

HRP-conjugated antibody. For the GST-pulldowns shown in Fig. 2a. 50 mM HEPES pH 7.4 was used instead of 50 mM Tris pH 7.4.

**Comparison of internal and external levels of S by flow cytometry**. U2OS cells were seeded at a density of $2 \times 10^4$ cells/cm$^2$ in T75 flasks in culture medium in a humidified incubator at 37 °C with 5% CO$_2$. Twenty-four hours after seeding, cells were transfected with 15 μg of plasmid DNA encoding different N-terminally HA-tagged S cytoplasmic tail mutants, and C-terminally FLAG-tagged M where specified, using PEI. Twenty-four hours after transfection, cells were washed once in EDTA solution and dissociated from the flask in accutase (Sigma) for 2 min at 37 °C. Cells were washed once in ice cold FACS buffer (2% FCS in PBS) by resuspension and centrifugation at $300 \times g$ for 5 min at 4 °C. The supernatant was removed and ~10$^6$ cells were resuspended in FACS buffer containing an anti-HA AF488 conjugate (1:1000, BioLegend, 901509) and an eFluor 780 fixable viability dye (Thermo Fischer Scientific, 65-0865-14). Cells were incubated on ice, in darkness for 30 min. Cells were washed 3 times in FACS buffer and incubated in Cyto-Fast Fix/Perm Buffer (BioLegend, 426803) for 20 min at room temperature. Cells were washed once in Cyto-Fast Wash Buffer and incubated in Cyto-Fast Wash Buffer containing an anti-HA AF647 conjugate (1:1000, BioLegend, 682404) and an anti-FLAG AF549 conjugate (for cells expressing FLAG-tagged M; 1:1000, Biolegend, 637314) for 20 min at room temperature. Cells were washed twice with Cyto-Fast Wash Buffer and resuspended in FACS buffer. Cells were strained using a 100 μm filter prior to analysis on an LSRII flow cytometer (BD Biosciences). Data were analysed using FlowJo v10 in which singlets were gated according to forward and side scatter profiles, dead cells were excluded using the viability dye and non-transfected cells were excluded based on their low AF488 and 647 signals. Single colour controls were used to conduct compensation. Gating parameters are shown in Supplementary Fig. 4. FlowJo v10 was used to calculate medians and perform chi-squared tests of their differences (Supplementary Data 3).

**Assay of endocytosis of S by flow cytometry**. Cells were seeded, transfected and resuspended as described above. The antibody uptake assay was based on the use of quenching to distinguish internal from external material[68]. Approximately 10$^6$ cells were resuspended in complete medium containing an anti-HA AF488 conjugate (1:1000) and incubated at 37 °C for 40 min. The cells were washed twice with ice cold FACS buffer and incubated with an anti-AF488 antibody (1:67, A-11094, Thermo Fischer Scientific; to quench non-internalised anti-HA AF488 conjugate), an anti-mouse AF647 antibody (1:300, Thermo Fischer Scientific, A31571; to relabel the non-internalised anti-HA conjugate) and an eFluor 780 fixable viability dye. Cells were washed three times in ice cold FACS buffer, fixed in 4% paraformaldehyde (PFA) for 20 min and washed a further two times in FACS buffer. Cells were strained and analysed as described above. Internalised anti-HA AF488 conjugate was inaccessible to quenching and therefore any associated AF488 signal was equated to levels of internalised S. Conversely, only anti-HA AF488 conjugate at the cell surface was accessible by the anti-mouse AF647 secondary antibody and therefore any associated AF647 signal was equated to levels of non-internalised S.

**Comparison of internal and external levels of S by immunofluorescence**. U2OS cells were seeded at a density of $2 \times 10^4$ cells/cm$^2$ in six-well plates in culture medium in a humidified incubator at 37 °C with 5% CO$_2$. Twenty-four hours after seeding, cells were transfected with 1–2 μg of plasmid DNA encoding different N-terminally HA-tagged S cytoplasmic tail mutants using PEI. Twenty-four hours after transfection, cells were washed once in EDTA solution and dissociated from the flask in trypsin for 2 min at 37 °C and seeded onto coated microscope slides (Hendley-Essex) in culture medium in a humidified incubator at 37 °C with 5% CO$_2$. 24 h after seeding, cells were washed with PBS and fixed in 4% PFA in PBS for 20 min at room temperature. Cells were incubated in an anti-HA AF647 conjugate (1:1000) diluted in 20% FCS in PBS for 20 min in order to label HA-tagged S at the surface of cells. Cells were washed three times in PBS and permeabilised in 10% Triton X-100 in PBS for 10 min. Cells were subsequently washed twice in PBS and incubated in blocking buffer (20% FCS, 1% Tween-20 in PBS) for 1 h. Cells were incubated with an anti-calreticulin antibody (1:200, Abcam, ab2907) or anti-calnexin antibody (1:1000, Abcam, ab22595), diluted in blocking buffer for 1 h in order to counterstain the ER. Cells were washed twice in PBS, incubated in blocking buffer for 10 min and washed twice again in PBS. Cells were incubated for 1 h in darkness with an anti-rabbit AF555 secondary antibody (1:300, Thermo Fischer Scientific, A31572) and an anti-HA AF488 conjugate (1:1000) diluted in blocking buffer with the latter intended to label HA-tagged S associated with intracellular membranes (as well as available external HA-S epitopes). Alternatively, when assaying the effect of co-transfection with M-FLAG, cells were incubated with an anti-rabbit AF405 secondary antibody (1:300, Abcam, ab175649) and an anti-FLAG AF594 conjugate (1:1000, Biolegend, 637314). Cells were washed twice in PBS, incubated in blocking buffer for 10 min and washed twice again in PBS. Cells were mounted in Vectashield (Vector Laboratories) prior to application of a coverslip which was sealed using nail vanish. Slides were imaged using a Leica TCS SP8 confocal microscope using Leica Application Suite X.

**S protein uptake assay**. U2OS cells were transfected with plasmids encoding N-terminally HA-tagged S and variants, or HA-tagged CPD, and seeded onto

microscope slides as described above. Forty-eight hours after transfection, cells were incubated in ice cold media and slides were placed on ice for 15 min. An anti-HA antibody (1:300, Roche, 3F10) diluted in ice cold media was added and cells were incubated on ice for 30 min. Cells were washed twice with ice cold PBS and subject to a chase in which warm media was added and cells were incubated in a humidified incubator at 37 °C with 5% $CO_2$ for 1 h. Cells were washed, fixed and blocked with 20% FCS in PBS as described above. Cells were incubated with an anti-rat AF647 secondary antibody (1:300, Abcam, ab150154) diluted in 20% FCS in PBS to label non-internalised anti-HA antibody under non-permeabilising conditions at room temperature for 20 min. Cells were washed three times in PBS and permeabilised and blocked as described above. Cells were incubated with an anti-SNX27 antibody (1:300, Abcam, ab77799) in blocking buffer for 1 h in order to counterstain endosomes. Cells were washed twice in PBS, incubated in blocking buffer for 10 min and washed twice again in PBS. Cells were incubated for 1 h in darkness with an anti-mouse AF555 secondary antibody (1:300, Thermo Fischer Scientific, A31571) and an anti-rat AF488 secondary antibody (1:300, Thermo Fischer Scientific, A21208) diluted in blocking buffer with the latter intended to stain internalised anti-HA antibody. Cells were washed, mounted and imaged as described above.

**Cell-cell fusion assay**. Acceptor and donor cells were seeded at 70% confluency in a 24-well plate 16 h prior transfection[46]. Donor 293T cells were co-transfected with 1.5 µg of plasmids encoding different untagged S mutants and 0.6 µg of pmCherry-N1 using 6 µL of Fugene 6 following the manufacturer's instructions (Promega). Acceptor Vero cells expressing human ACE2[46], were treated with CellTracker™ Green CMFDA (5-chloromethylfluorescein diacetate, Thermo Scientific) for 30 min. Donor cells were then detached 5 h post transfection, mixed together with the green-labelled acceptor cells and plated in a 12-well plate. Cell-cell fusion events were measured using an IncuCyte (Sartorious) and determined as the proportion of merged area to green area over time. Data were collected using IncuCyte software v2019B and plotted with GraphPad Prism v8.

**Reporting summary**. Further information on research design is available in the Nature Research Reporting Summary linked to this article.

## Data availability

Mass spectrometry data generated in this study have been deposited to the ProteomeXchange Consortium via the PRIDE[64] partner repository with the dataset identifier "PXD022215", and the processed data are summarised in Supplementary Data 1. All other reagents and relevant data supporting the key findings of this study are available within the article and its Supplementary Information files or from the corresponding author upon reasonable request. Source data are provided with this paper.

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

## Acknowledgements

We thank Manu Hegde for comments on the manuscript, John James and Natalya Leneva for reagents, Maria Daly for help with flow cytometry, Mark Skehel for help with the mass-spectrometry data, and David Owen for advice on porcine diarrhea. Funding was from the Medical Research Council, as part of United Kingdom Research and Innovation (also known as UK Research and Innovation), (File reference number MC_U105178783).

## Author contributions

S.M. devised, and J.C.O., L.G.W. and S.M. planned, the study. J.C.O. and L.G.W carried out all the biochemistry, immunofluorescence and flow cytometry. S.L.M. performed the mass spectrometry, and processed the raw data. G.P and L.C.J. devised and performed the cell fusion assay. S.M. wrote the manuscript.

## Competing interests

The authors declare no competing interests.
