## [Peer Review File · Nature Communications]

REVIEWERS' COMMENTS

Reviewer #1 (Remarks to the Author):

The Authors have addressed the concerns raised in my previous review by acknowledging the finding by Carolyn Machamer concerning the suboptimal binding of COPI by the Spike protein of SARS-CoV and by co-expressing the M protein with the Spike protein of SARS-CoV-2. Surprisingly while, as expected, the wt S protein is retained by and colocalizes with M protein in the Golgi complex, the H1271K and the T1273A S mutants neither are affected by the M protein nor they colocalize with it: have they lost the ability to interact with M?

Reviewer #3 (Remarks to the Author):

The work by Cattin-Ortolá, Welch and colleagues has been extensively expanded with new experimental data that further support their conclusions. The authors have responded in depth to our comments and have added substantial and important new data that addressed all our questions and concerns.

Importantly, the authors have now shown the direct nature of Spike(S):COPI interaction by recapitulating the binding between purified recombinant β -COP (residues 1-304) and recombinant S tail. This result strengthens the previous immunoprecipitation experiments where recombinant S tail was used to immunoprecipitate COPI from HEK293T cell lysates (hence indirect association couldn't be excluded). Moreover, the manuscript contains new data quantifying the effects that S tail mutations have on the binding to COPI/COPII. Further light microscopy-based imaging has also clarified the trafficking differences between wild type and mutant Spike proteins.

By exploring the role of a suboptimal COPI binding motif in the cytosolic tail of SARS-CoV-2 Spike the authors report a cellular mechanism that could account for increased surface accumulation of Spike in infected cells. This process, in combination with other aspects of SARS-CoV-2 biology, might account for the fact that SARS-CoV-2 induces cell fusion more efficiently than SARS-CoV. SARS-CoV-2 S trafficking and presentation to the cells surface is relevant for understanding the immunogenicity of S-based vaccines and exploring new antiviral approaches. Overall, we thank the authors for their work, and we recommend the publication of the manuscript in Nature Communications.

Reviewer #4 (Remarks to the Author):

The manuscript "Sequences in the cytoplasmic tail of SARS-CoV-2 Spike facilitate expression at the cell surface and syncytia formation" by Cattin-Ortola and colleagues investigates potential mechanism in which Spike accumulates at the cell surface. Their findings are unique and provide an important mechanistic advance in understanding syncytia formation by SARS-CoV-2. They suggest that suboptimal COPI-binding residues in the Spike protein allows for leakage from the Golgi which

ultimately allows for plasma membrane accumulation and cell-cell fusion.

In order to show that mutations in the Spike protein COPI binding site differentially affect syncytia formation, they perform an acceptor-donor experiment where human 293T cells transfected with spike protein are co-cultured with Vero cells expressing human ACE2. The previous reviewer had expressed concern regarding the physiological relevance of this model and editor has solicited our opinion on the matter.

Generally speaking, the acceptor-donor syncytia formation system is used to assess the fusogenicity of the different spikes (WT vs mutants) and provides a quantifiable comparison and is not intended to replicate physiological syncytia formation. The transfection of the donor 293T cells with Spike is suitable because they do not express endogenous ACE2, thus preventing donor-donor fusion. The primary reason to use Vero cells as acceptors is for their endogenous ACE2 expression, albeit monkey ACE2, which reduced variations in intra/intercellular ACE2 expression. Endogenous Vero ACE2 expression by itself can induce cell-cell fusion with acceptor cells expressing spike. This would have been sufficient for the author's intended characterization of mutant spike proteins. Thus, it is curious that the authors used Vero cells that are also expressing human ACE2. If their goal is to demonstrate spike mediated fusion specifically with the human ACE2 then using human cell lines like A549 or U2OS cells transduced with ACE2 as the acceptor cells would have been acceptable. However, as the author state that they are just assessing the degree of cell-cell fusion elicited by the spike mutants, the system used provides relevant information within its own confines.

Response to reviewers' comments.

We are very pleased that the reviewers felt that we had done a good job in addressing their comments and concerns, and hence they were happy to recommend publication. To address the remaining concerns of Reviewers #1 and #4, we have, as requested, added further discussion in the text about the interaction between S and M, and also discussed the caveats of the cell fusion assay. We have also addressed all of the editorial requests, as outlined in our responses in the Author Checklist, and have ensured that the manuscript complies with the policies and formatting requirements of Nature Communications.

Reviewer #1 (Remarks to the Author):

The Authors have addressed the concerns raised in my previous review by acknowledging the finding by Carolyn Machamer concerning the suboptimal binding of COPI by the Spike protein of SARS-CoV and by co-expressing the M protein with the Spike protein of SARS-CoV-2. Surprisingly while, as expected, the wt S protein is retained by and colocalizes with M protein in the Golgi complex, the H1271K and the T1273A S mutants neither are affected by the M protein nor they colocalize with it: have they lost the ability to interact with M?

We have added further discussion in the results to address this issue, and the relevant section now reads:

"Immunofluorescence shows that these variants still have the substantial ER staining seen in the absence of M (Supplementary Fig. 3c). In these cases, M does not accumulate in the ER as well, but rather is still localised to the Golgi. This is consistent with previous studies with other coronaviruses that found that S and M do not associate in the ER where they are both synthesised, but rather they only assemble after they have accumulated at the site of virion budding in the early Golgi^{20,42,43}. Thus, even in the presence of M, the COPII binding site in S is required for exit from the ER, and an optimised COPI binding site in S can be recognised so as to reduce transport to the surface."

Reviewer #4 (Remarks to the Author):

Thus, it is curious that the authors used Vero cells that are also expressing human ACE2. If their goal is to demonstrate spike mediated fusion specifically with the human ACE2 then using human cell lines like A549 or U2OS cells transduced with ACE2 as the acceptor cells would have been acceptable. However, as the author state that they are just assessing the degree of cell-cell fusion elicited by the spike mutants, the system used provides relevant information within its own confines.

We have added further discussion in the results to make this caveat clear, and the relevant section now reads:

"It should be noted that this assay was performed using as the fusion target monkey (Vero) cells overexpressing human ACE2. This will increase the susceptibility of the cells to S-mediated fusion, and hence it is striking that the H1271K mutant still reduces fusion, but it also means that caution is needed in extrapolating these findings to other cell types. Nonetheless, the results clearly indicate that, at least in this context, the sub-optimal COPI binding site enhances the ability of S to form syncytia."